# Understanding Physicochemical Mechanisms of Sequential Infiltration Synthesis toward Rational Process Design for Uniform Incorporation of Metal Oxides

**DOI:** 10.3390/s22166132

**Published:** 2022-08-16

**Authors:** Jiwoong Ham, Minkyung Ko, Boyun Choi, Hyeong-U Kim, Nari Jeon

**Affiliations:** 1Department of Materials Science and Engineering, Chungnam National University, Daejeon 34134, Korea; 2Department of Plasma Engineering, Korea Institute of Machinery & Materials (KIMM), Daejeon 34103, Korea

**Keywords:** sequential infiltration synthesis (SIS), in situ fourier transform infrared spectroscopy, aluminum oxides, growth mechanism, atomic layer deposition (ALD)

## Abstract

Sequential infiltration synthesis (SIS) is a novel technique for fabricating organic–inorganic hybrid materials and porous inorganic materials by leveraging the diffusion of gas-phase precursors into a polymer matrix and chemical reactions between the precursors to synthesize inorganic materials therein. This study aims to obtain a fundamental understanding of the physicochemical mechanisms behind SIS, from which the SIS processing conditions are rationally designed to obtain precise control over the distribution of metal oxides. Herein, in situ FTIR spectroscopy was correlated with various ex situ characterization techniques to study a model system involving the growth of aluminum oxides in poly(methyl methacrylate) using trimethyl aluminum (TMA) and water as the metal precursor and co-reactant, respectively. We identified the prominent chemical states of the sorbed TMA precursors: (1) freely diffusing precursors, (2) weakly bound precursors, and (3) precursors strongly bonded to pre-existing oxide clusters and studied how their relative contributions to oxide formation vary in relation to the changes in the rate-limiting step under different growth conditions. Finally, we demonstrate that uniform incorporation of metal oxide is realized by a rational design of processing conditions, by which the major chemical species contributing to oxide formation is modulated.

## 1. Introduction

Sequential infiltration synthesis (SIS) is an emerging synthetic route used to fabricate organic-inorganic hybrid materials by exploiting the newly discovered phenomenon involving the diffusion of gaseous precursors into a polymer and chemical reactions with co-reactants within the polymer [1,2,3,4]. Several papers in the early days of SIS research reported interesting properties of SIS-derived materials suitable for diverse applications, such as the high toughness of spider silks [5], etch resistance of photoresists [6,7], and surface area of an anode layer in dye-sensitized solar cells [8]. For the past years, the potential application areas for SIS materials have expanded further, encompassing various domains, such as triboelectricity [9], photovoltaics [10,11,12], antireflection [13,14], filtration [15,16,17], sensors [18], and memories [19]. Along with the expansion of SIS applications, extensive efforts have been devoted to understanding the process kinetics with the aid of various characterization techniques.

Common in situ characterization tools equipped with an atomic layer deposition (ALD) reactor, such as quartz crystal microgravimetry (QCM), Fourier transform infrared (FTIR) spectroscopy, and spectroscopic ellipsometry (SE), are well-suited to monitor SIS processes [20,21]. QCM is sensitive to mass changes that occur as a response to the precursor dose and purge and has been utilized to optimize the growth temperature at which the mass gain per pulse is maximized [22]. The in-situ SE has enabled monitoring of the swelling and deswelling of the polymer film in a response to the infiltration and purge of trimethyl aluminum (TMA) in polyethersulfone and poly (methyl methacrylate) (PMMA) [23,24,25]. Additionally, in situ X-ray photoelectron spectroscopy (XPS) studies have been used to identify optimal process parameters, such as the precursor hold time and temperature [26]. Various ex-situ characterization methods have also been used. For example, time-of-flight secondary ion mass spectroscopy analysis was used to characterize the non-uniform distribution of aluminum near the polymer surface and polymer/substrate interface depending on the thickness of polymer films [27]. Synchrotron X-ray based techniques have identified as-grown oxides as three dimensional (3D) arrays of discrete atom number clusters [28]. Among the various in situ and ex situ characterization methods, the in situ FTIR spectroscopy characterization technique, intensively used in this study, exhibits distinctive merits, such as the capability of real-time observation of various chemical reactions occurring in the IR beam path (including both a vapor and polymer phase) and quantification of the observed chemical reactions.

The main mechanism of precursor infiltration into a polymer matrix includes the following steps: (1) sorption of the precursors into the polymer, (2) diffusion of the precursors in the polymer, and (3) entrapment of the precursors in the polymer [2,27,29,30]. The entrapment of the precursor is due to the formation of reversible complexes between the sorbed precursor molecules and specific functional groups of the polymer. Weisbord et al., employed a combined approach of density function theory (DFT) calculations and in situ QCM measurements to ascertain the temperature point at which the rate of the association and dissociation reactions of precursor-polymer complexes are equal [31]. They proposed that this balance point is the optimum growth temperature for maximum mass gain. Recently, a reaction-diffusion model was developed that described temporal changes in the diffusivity of precursors owing to the immobilization of precursors, which served as diffusion barriers [32].

Most of the previous research focuses on understanding the physicochemical process involved in the SIS of a “single” cycle. Ideally, a single SIS cycle may be sufficient for maximum oxide formation for a given thickness of a polymer film, provided that a sufficient time is allowed for precursor diffusion. However, the use of multiple SIS cycles provides a more practical approach for the efficient growth of inorganic materials in the polymer matrix within a reduced processing time. The multiple SIS cycles will complicate the physicochemical processes, especially in the later SIS cycles, because the precursors will diffuse to the organic-inorganic hybrid matrix instead of the pristine polymer, as they would in the first SIS cycle. Additionally, the distribution of inorganic material in a polymer was reported to be non-uniform along the direction of the film thickness [30,33,34]. However, theoretical approaches to understand the non-uniform distribution or process design to achieve a uniform or tailored distribution of inorganic materials in a polymer phase have rarely been reported.

The goal of this work is to obtain a fundamental understanding of the physicochemical mechanisms underlying SIS, based on which SIS processing conditions are rationally designed to achieve precise control over the distribution of metal oxides. To this end, we attempted to answer the following questions under different SIS conditions: (1) What is the rate-limiting step: formation of the precursor-polymer complex vs. precursor diffusion? (2) What are the primary chemical species that contribute the most to the oxide formation?

Herein, a model system involving the growth of aluminum oxides in poly(methyl methacrylate) was developed using trimethyl aluminum and water as the metal precursor and co-reactant, respectively. Finally, we demonstrate that the uniform incorporation of aluminum oxide is realized with a rational process design, in which the major chemical species and preferred growth sites for oxide formation are modulated.

## 2. Materials and Methods

### 2.1. Sample Preparation

PMMA (Sigma-Aldrich, Waltham, MA, USA, average M_w_~15,000) was dissolved in toluene (Sigma-Aldrich, anhydrous, 99.8%) at 3.5 wt.% by stirring the solution for at least 24 h at 50 °C. The PMMA-toluene solution was then filtered using a polytetrafluoroethylene (PTFE) syringe filter with a 0.4 μm pore size. The filtered solution was used for spin coating a PMMA film on a Si (100) substrate with an ~2 nm thick native SiO_2_ layer. The spin-coating speed and duration were 3000 rpm and 30 s, respectively. The spin-cast samples were annealed on a hot plate at 80 °C for 5 min in air to promote the evaporation of toluene. The average thickness of the as-spun PMMA films was ~58 nm. For the in situ FTIR spectroscopy experiments, a PMMA film was prepared on a 100 nm thick Au layer, which was deposited on a Si substrate by e-beam evaporation.

### 2.2. SIS

The SIS was performed with a laminar-flow thermal ALD reactor (Daeki Hightech, Mado-Myeon, Korea) designed for a single 4” wafer. During SIS processes, a 5 SCCM of Ar flow (99.995% purity) was constantly provided to the reactor, which yielded a base pressure of 0.2 Torr in the reactor. The gas lines for the precursor delivery were maintained at 80 °C. TMA (EZChem) and H_2_O (Sigma-Aldrich) were maintained at room temperature. A single SIS cycle was composed of a TMA half-cycle and a H_2_O half-cycle, with each half-cycle containing three steps: precursor dose, exposure, and purge. Immediately before the dose step, the main chamber and pump were disconnected by closing the gate valve located downstream of the reactor. The Ar flow was maintained, the downstream gas line was disconnected from the pump during the dose and exposure step, and the pressure in the chamber was gradually increased to the level of atmospheric pressure. At the start of the purge step, the valve between the chamber and pump was opened again, and therefore, the chamber pressure rapidly decreased to the base level within a few seconds. The precursor dose, exposure, and purge time at each characterization step are summarized in the Appendix A.

### 2.3. In Situ FTIR Spectroscopy

In situ FTIR spectroscopy measurements were conducted using an FTIR spectrometer (Bruker, Invenio-R) with a mercury–cadmium–telluride (MCT) detector in the reflection mode. In the experimental setup (Appendix A), an IR beam was focused on a sample located at the center of the ALD reactor with an incidence angle of ~75° via a 90° off-axis parabolic mirror. The IR beam was reflected from the sample and collected using a MCT detector. The measured sample had an ~100 nm thick Au layer below the PMMA layer to enhance IR reflectivity. All the samples used for in situ FTIR spectroscopy measurements in this study have the structure of PMMA/Au/Si unless otherwise noted. The absorbance spectrum of the Au-coated Si sample without the PMMA layer (i.e., Au/Si) was used as the background spectrum, and the difference spectrum of the PMMA/Au/Si sample was collected. The absorbance spectrum of the pristine PMMA/Au/Si sample, obtained immediately before each SIS experiment, was used as a background spectrum to calculate the difference spectra during the SIS experiments. Time-lapse spectra with 8 cm^−1^ spectral resolution were collected at every ~5 s. An IR beam of ~3.5 mm size was focused on the PMMA/Au/Si sample via a 90° off-axis parabolic mirror. The door of the ALD chamber was customized to have two IR windows of CaF_2_ with an average IR transmissivity of 90%.

### 2.4. Other Characterization Methods

XPS depth profiles were obtained using a Thermo Fisher Scientific K-Alpha^+^ system under the following conditions: 3 kV Ar^+^ sputtering voltage, Al Kα (1487 eV) X-ray source of 200 mm size, and 1mm^2^ rater area. The binding energies (BEs) for Al 2p and O 1s high resolution (HR)XPS were calibrated using the BE of the adventitious C signal and the signals from C-C/C-H carbons of the PMMA component. The Voigt line shape was used for the deconvolution of HRXPS. A spectroscopic ellipsometer (M-2000, J. A. Wollam) was used for the measurement of the PMMA-AlOx composite films. The Cauchy model was used with the film thickness and refractive index as the fitting parameters.

## 3. Results

### 3.1. In Situ FTIR Spectroscopy

The absorbance spectrum of PMMA/Au/Si mainly contained the IR-active vibrational modes of the functional groups in the PMMA layer: C=O stretching at 1738 cm^−1^, CH bending of CH_3_ at 1443 cm^−1^, CH bending of CH_2_ at 1483 cm^−1^, C-O-R stretching at 1269 and 1151 cm^−1^, symmetric CH stretching of CH_3_ at 2951 cm^−1^, and CH stretching of CH_2_ at 2997 cm^−1^ (Figure 1a) [29]. The absorbance spectra of the samples during the SIS process were collected at every ~5 s. The absorbance spectrum of the pristine sample prior to precursor exposure was used as the background spectrum. In the in-situ difference spectra, temporal changes in the IR bands were observed that have originated from the gas-phase TMA precursors as well as the IR-active functional groups of PMMA during the SIS process. Figure 1b shows the absorbance spectra collected at different times during the TMA dose/exposure step. Both the stretching and bending modes of the gas-phase TMA molecules were observed in the range of 3140–2780 cm^−1^ and 1380–1150 cm^−1^, respectively. As TMA molecules in the gas phase were sorbed in the PMMA phase, TMA-PMMA complexes were formed via a Lewis acid-base reaction between TMA and the C=O functional group of PMMA [29,35,36]. The vibrational frequency of the C=O mode in the complex was redshifted; therefore, the formation of the complex was manifested with the reduction of a pristine C=O band (1738 cm^−1^) and the growth of a shifted C=O band (1670 cm^−1^) [29,35,36].

In the absorbance spectra, the absorbance (A) of the shifted C=O band can be used as a parameter to estimate the concentration (C) of the TMA-PMMA complexes according to the Beer–Lambert law:(1)A=εCl

In Equation (1), ε is the molar extinction coefficient of the C=O bond weakly bound to TMA, and l is the film thickness. Assuming ε and l as constants, changes in A as a function of time during the SIS process describe the kinetics of association and dissociation processes of the TMA-PMMA complexes. For example, Figure 1c shows A of the shifted C=O band during the following steps: TMA dose (2 s); TMA exposure (1800 s); TMA purge (1200 s); H_2_O dose (2 s); H_2_O exposure (600 s). The A rapidly increased and saturated during the TMA dose–exposure step and decreased in the TMA purge step. Upon H_2_O dose and exposure, A rapidly decreased to ~0 owing to the formation of Al-O covalent bonds [29].

### 3.2. Precursor Sorption

The saturated A of the shifted C=O band ($A_{shifted\;C=O}$) for a given condition was used to estimate the equilibrium concentration of TMA weakly bound to the C=O groups of the PMMA matrix in the temperature range of 80–200 °C. Therefore, the TMA exposure time was chosen to be sufficiently long so that the $A_{shifted\;C=O}$ value was saturated at the end of the TMA exposure step. Figure 2 shows the natural log of $A_{shifted\;C=O}$ as a function of the inverse of the growth temperature. The equilibrium concentration (C) of TMA molecules sorbed in the PMMA matrix at different temperatures is related to the enthalpy of sorption (ΔH): [30]
(2)$$C\propto \exp\left(-\frac{\Delta H}{kT}\right)$$

Equation (2) is derived from the Van’t Hoff relation for sorption. There are two temperature regions with different ΔH values: −0.03 eV in a low-temperature region below 110 °C and −0.42 eV in a high-temperature region above 120 °C. The ∆H value of −0.03 eV in the low-temperature region is the same as that measured by SE experiments as reported by Leng et al. [30]. The agreement between the FTIR and SE results is not necessarily guaranteed because the operating principles for the characterization techniques are different. While FTIR spectroscopy probes TMA molecules that form weak complexes with PMMA, SE detects all TMA molecules that contribute to the increase in the density of a PMMA film without considering their chemical status (i.e., differentiation between a weakly binding state and a freely diffusing state does not occur in SE). Considering these differences, the excellent agreement of ΔH results obtained from the two different characterization methods in the low-temperature region below 110 °C implies the following: (1) most of the TMA sorbed in the PMMA matrix is present in the form of TMA-PMMA complexes, and/or (2) the ratio between the TMA molecules that form the weak complex with PMMA and those that exist in a freely diffusing state does not significantly change with an increase in temperature in the limited temperature range below 110 °C.

The sorption became increasingly exothermic at temperatures above 120 °C. Leng et al. reported a similar trend of a more exothermic nature at higher temperatures based on SE measurements; however, the transition temperature (90 °C) and ΔH value (−0.33 eV) in the high-temperature region in that study were slightly different from the present study [30]. The sudden shift in the behavior of PMMA toward a more exothermic nature at temperatures above the glass temperature (T_g_) is related to the formation of PMMA–TMA hybrid material, which better mixes with TMA molecules [30]. The differences in the transition temperatures arise because T_g_ depends on various factors, such as molecular weight and tacticity. The lower value of ∆H in this study (−0.42 eV), compared to that in the previous SE study (−0.33 eV), may imply that the concentration of bound TMA molecules decreases more rapidly compared to that of the unbound TMA molecules with an increase in temperature in the temperature range investigated [31]. Finally, a caveat needs to be considered, as explained here. The PMMA film may expand as a response to the temperature increase and precursor sorption. However, the PMMA film is expected to expand mostly along the thickness, because the lateral dimension of the film is constrained by that of the underlying substrate. Therefore, we did not calibrate for the effect of the volume expansion on the FTIR measurement, that is, the possible decrease in the number of C=O groups in a fixed-size IR beam. Therefore, we propose that a correlation of in situ FTIR spectroscopy (after correction for the volume expansion) and SE may allow for the precise differentiation between the concentration of (1) the sorbed TMA molecules that freely diffuse in to PMMA and (2) the sorbed TMA molecules that form weak complexes with PMMA at different temperatures.

### 3.3. Precursor Diffusion

Changes in $A_{shifted\;C=O}$ as a function of time t during different SIS steps were used to understand the kinetics of the TMA-PMMA complex formation. In the TMA dose/exposure step, TMA molecules were sorbed and diffused into the PMMA phase to form TMA-PMMA complexes. A portion of the TMA-PMMA complexes was dissociated so that: (1) the released TMA molecules could further diffuse into the PMMA matrix to form new TMA-PMMA complexes, and (2) newly sorbed, freely diffusing TMA molecules could be bound to the released C=O groups to form TMA-PMMA complexes near the PMMA surface (inset of Figure 3a). Consequently, the temporal changes in $A_{shifted\;C=O}$ are related to the kinetics of the formation of TMA-PMMA complexes as well as the diffusion of unbound TMA molecules. The changes in $A_{shifted\;C=O}$ as a function of time during the TMA dose/exposure step (Figure 3a) were fitted with a mono-exponential function as follows:(3)$$A=A_{0}+B\exp\left\{-\frac{\left(t-t_{0}\right)}{\tau_{e}}\right\}$$

In Equation (3), $A_{0}$, B, $t_{0}$, and $\tau_{e}$ are the initial absorbance prior to the TMA dose step, saturated absorbance during the TMA exposure step, time at the start of the TMA dose step, and effective time constant related to the absorbance saturation, respectively. The effective time constant *τ_e_* is a parameter used to compare the kinetics for the complex association convoluted with the TMA in-diffusion under different conditions [29,37,38]. In this study, the measured *τ*_e_ values at 80 °C are similar to those measured by in situ FTIR spectroscopy under similar SIS conditions in terms of the PMMA thickness and growth temperatures as reported in literature [36]. However, it is much faster compared to the values measured by ex situ SE [30]. For example, the *τ_e_* measured at 100 °C in this study was ~18 s, whereas *τ_e_* from the ex-situ SE study at the same temperature for a similar PMMA thickness (80 nm) was ~120 s [30,37]. However, a direct comparison between the in situ and ex situ measurements cannot be made because the results of ex situ measurements include the effects of the TMA purge step, in which a portion of the sorbed TMA in PMMA diffuses out from the PMMA matrix. The temporal changes in $A_{shifted\;C=O}$ can also be used to extract a diffusion coefficient, assuming that TMA diffusion follows the behavior of Fickian diffusion (Appendix A). However, the analysis of the diffusion coefficient was only valid at low temperatures (<90 °C) because the saturation of $A_{shifted\;C=O}$ occurred more rapidly than the time difference between two consecutive FTIR measurements at higher temperatures.

Figure 3b shows the effective time constants measured in the TMA purge step at different temperatures. The purge step was performed after the TMA exposure step, with an exposure time of 10 × *τ_e_* for a given temperature, as described above. A long exposure time was selected to fully saturate the PMMA films with TMA molecules at a given temperature. At a growth temperature of 110 °C or higher, the temporal changes in $A_{shifted\;C=O}$ were well fitted with a mono-exponential function. In the high-temperature range, Equation (3) was used with the time constant *τ_p_* instead of *τ_e_*. In contrast, a bi-exponential function (Equation (4)) with two time constants, *τ*_*p*1_ and *τ*_*p*2_, was necessary to fit the temporal changes in $A_{shifted\;C=O}$ at low growth temperatures (≤100 °C).
(4)$$A=A_{0}+B_{1}exp\left\{-\frac{\left(t-t_{0}\right)}{\tau_{p1}}\right\}+B_{2}exp\left\{-\frac{\left(t-t_{0}\right)}{\tau_{p2}}\right\}$$

The presence of an additional, very long time constant (*τ_p_*_2_) in the low-temperature region may imply that the role of complex association/dissociation in comparison to that of out-diffusion is more significant in determining the total kinetics of the purge step in the low-temperature region. Therefore, two possible scenarios for the presence of extremely long time constants at low temperatures are proposed. Firstly, TMA-PMMA complexes with a slightly stronger binding may exist. Elam et al., suggested that the stable Al-O covalent bond is later formed from the weakly-bound TMA-PMMA complex with slower kinetics [38,39]. Although the time scale for the complete formation of Al-O covalent bonding suggested in previous reports was much longer than the measured time constant of ~1700 s at 80 °C in this study, it may be possible that a distribution of binding energies among a number of TMA-PMMA complexes evolved during the prolonged TMA exposure step at low temperatures. The dissociation kinetics of the complexes with stronger binding are slower than those of the complexes with weaker binding. Secondly, the diffusion kinetics for the released, unbound TMA molecules is slower at a low temperature, and therefore, TMA molecules tend to stay in the PMMA phase longer. The increased residence time of TMA molecules in the PMMA matrix increased their chances of being re-bound to the C=O functional groups before they were completely evacuated from the PMMA matrix.

The TMA dose/exposure step was followed by a short TMA purge step of 2 s and a H_2_O half-cycle (0.5 s dose, 600 s exposure, and 120 s purge). The short purge time of TMA was designed to capture as many sorbed TMA molecules as possible by reacting them with H_2_O in the subsequent H_2_O dose and exposure steps. The TMA purge step is inevitable because the TMA precursors in the vapor phase must be evacuated before the introduction of H_2_O into the chamber. The out-diffusion of TMA from the PMMA matrix in the H_2_O half-cycle may be possible; however, the in-diffusion of H_2_O into the PMMA matrix is much faster so that most of the sorbed TMA will react with H_2_O to form permanent Al-O bonds in PMMA [40]. We therefore assumed that all the TMA molecules remaining in PMMA at the end of the TMA exposure step instantaneously reacted with H_2_O upon the H_2_O dose step. Figure 4a compares the $A_{shifted\;C=O}$ value at the end of the TMA exposure step with the film swelling percentage measured by ex situ SE at different temperatures. The swelling of the polymer film after SIS is proportional to the amount of inorganic materials synthesized in the polymer film, provided they are formed uniformly throughout the polymer matrix [24]. The film swelling percentage and the absorbance $A_{shifted\;C=O}$ displayed a similar trend of continuous decrease with an increase in the growth temperature. This result indicates that the $A_{shifted\;C=O}$ value can serve as a comparable and indicative parameter for oxide formation under different SIS conditions when the precursor purge time is sufficiently short.

The similar Al_2_O_3_ SIS of 1 cycle was performed but with a much longer TMA purge step. The TMA purge time was chosen to be 10 times longer than the *τ_p_* determined for a given temperature. For temperatures of 100 °C or below, 10 × *τ_p_*_1_ (i.e., the time constant of a shorter time scale) was used for the TMA purge time. As shown in Figure 4b, $A_{shifted\;C=O}$ decreased rapidly at low temperatures, whereas the film swelling decreased gradually compared to $A_{shifted\;C=O}$ in the temperature range investigated. The faster decrease in $A_{shifted\;C=O}$ with the increase in temperature in the low-temperature region is associated with the TMA-PMMA complexes that are more easily dissociated by an increase in temperature. At high temperatures, most of the TMA molecules that were oxidized during the subsequent H_2_O half-cycle originated from those that were present in an unbound state in the PMMA matrix. In other words, the out-diffusion of TMA became a rate-limiting step compared with the association/dissociation of the complexes at high temperatures. Therefore, the slightly different trends of the swelling percentage and absorbance as a function of temperature are understood in terms of the difference in contribution between the bound TMA molecules and the unbound TMA molecules to the formation of aluminum oxides.

### 3.4. Uniform Distribution of Aluminum Oxides/Hydroxides

The SIS process at a lower temperature favors the efficient formation of aluminum oxides as long as the precursor exposure step is sufficiently long for the following reasons. First, the equilibrium concentration of TMA sorbed in the PMMA matrix was higher at lower temperatures. Second, the out-diffusion of TMA molecules to the vapor phase in the TMA purge step was slower at lower temperatures. However, efficient oxide formation at a reduced cycle number (and therefore, an economical usage of precursors) is not merely related to the uniform distribution of the oxide density along the direction of the film thickness. Therefore, the possibility that both efficient oxide formation and uniform oxide distribution may be achieved via the modulation of the SIS growth conditions was investigated.

We first performed a single cycle at 80 °C and evaluated the depth profile of the Al mole fraction (i.e., [Al]/([Al] + [C]). A TMA half-cycle constitutes 2 s dose, 1800 s exposure, and 600 s purge, and H_2_O half-cycle constitutes 2 s dose, 600 s exposure, and 600 s purge. As shown in Figure 5a, the Al distribution was uniform throughout the film thickness. The Al 2p HRXPS data collected at a different etch time of the sample confirmed the uniform chemical environments surrounding Al atoms with a major component of Al-OH bonding possessing a BE of ~75.1 eV (Figure 5b and Appendix A). The dominant presence of Al-OH rather than Al-O was expected because of the low growth temperature [41]. When five SIS cycles were conducted at 80 °C, the Al mole fraction substantially decreased from ~80 to ~50% in the near-surface region along the thickness direction (Figure 5a). The Al 2p HRXPS data collected in the near-surface region (Figure 5c) were deconvoluted into two peaks of Al-O bonding (~74.3 eV) and Al-OH bonding (~75.2 eV). However, below the near-surface region, the contribution of the Al-OH peak was mostly suppressed, and the spectra presented a single peak at ~75.0 eV (Figure 5c and Appendix A).

To achieve a uniform density of the Al mole fraction and efficient oxide formation, we performed SIS at two different temperatures: the 1st SIS cycle was performed at 80 °C, and the subsequent 4 cycles were performed at 130 °C. The 1st low-temperature step was designed to achieve an efficient loading of oxides/hydroxides, whereas the subsequent high-temperature steps promoted the uniform formation of oxides/hydroxides while preventing the formation of a dense surface layer that may act as a diffusion barrier for TMA molecules. As shown in Figure 5a, the Al mole fraction in the sample with the two-temperature scheme was uniform throughout the film. In addition, all the HRXPS spectra collected at different locations of the film were well fitted with a single component of Al-OH (~75.3 eV) (Figure 5d). This indicates that the chemical environment surrounding the Al atoms in this sample was similar to that in the sample subjected to a single cycle at 80 °C.

Figure 6a,b compare the absorbance spectra collected at the end of the TMA and H_2_O purge steps for the two samples prepared by the low-temperature 5-cycle SIS and the two-temperature 5-cycle SIS. IR bands were observed as follows: (1) pristine C=O mode at 1738 cm^−1^, (2) shifted C=O mode at 1670 cm^−1^, (3) surface -OH group in the range of 3000–3500 cm^−1^, and (4) surface -CH band at 2923 cm^−1^. In the low-temperature 5-cycle SIS spectra, small positive peaks observed at 1710 cm^−1^ and at 1540 cm^−1^, which appeared in the 1st H_2_O purge step, 2nd H_2_O purge step, and all subsequent steps of both TMA and H_2_O purge, originated from H_2_O present at a trace level. The incorporation of H_2_O molecules into the reactor chamber was inevitable owing to the inefficient removal of H_2_O molecules at a low temperature (80 °C), even with a long purge time (120 s) (Appendix A). In the spectra of the two-temperature 5-cycle SIS, a feature in the range of 1000–1700 cm^−1^ appeared in the 3rd cycle and later indicated the decrease of H_2_O molecules in the reaction chamber, owing to the increase in the growth temperature to 130 °C (Appendix A).

In the experiment of five-cycle SIS at 80 °C, $A_{shifted\;C=O}$ at the end of 1st and 2nd TMA exposure steps were ~3.1 × 10^−2^ a.u. and ~1.5 × 10^−2^ a.u., respectively. However, the $A_{shifted\;C=O}$ peak at the end of the 3rd and subsequent TMA exposure steps was largely suppressed below ~1.0 × 10^−3^ a.u., which is comparable to the noise level. The $A_{pristine\;C=O}$ peak at the end of the 1st and 2nd TMA exposure steps showed an overall trend similar to that of $A_{shifted\;C=O}$. However, $A_{pristine\;C=O}$ at the end of the 3rd and subsequent TMA exposure steps remained approximately constant at ~3 × 10^−3^ a.u. The constant and non-negligible value of $A_{pristine\;C=O}$ in the 3rd and subsequent cycles implies that the vibrational nature of some of the C=O groups in PMMA was permanently changed, which was likely due to the formation of AlO_x_H_y_ clusters. The overall changes in $A_{shifted\;C=O}$ and $A_{pristine\;C=O}$ at the end of each TMA exposure step with the increase in the cycle number indicated that the additional association of TMA to PMMA to form hybrid complexes was suppressed from the 3rd and subsequent cycles. The early termination of the TMA-PMMA complex formation is mainly due to the limited diffusion of TMA molecules in the hybrid film as the SIS cycle was repeated. The effective time constant in the 2nd TMA exposure step was longer than that in the 1st TMA exposure step, indicating a slower rate of diffusion in the 2nd exposure step (Appendix A). Additionally, the ex-situ SE measurement results showed that the expansion of the PMMA film was almost terminated after the 2nd cycle (Appendix A).

In the two-temperature SIS experiment, an increase in the $A_{pristine\;C=O}$ in each TMA exposure step was repeatedly observed for all 5 cycles, indicating the continued interaction between TMA and PMMA upon TMA exposure for all cycles. This trend was contrary to that observed in the single-temperature experiment at 80 °C, in which the association of TMA-PMMA complexes no longer occurred after the 2nd cycle. The $A_{shifted\;C=O}$ displayed a similar behavior to $A_{pristine\;C=O}$ until the 3rd cycle, but it had a negative value for the 4th and 5th cycles. The negative values of $A_{shifted\;C=O}$ in the later cycles are mainly artifacts, which originate from the background fitting and subtraction processes, as described in the Appendix A. After the 4th cycle $A_{shifted\;C=O}$ did not appear during the TMA exposure step, which indicated the nature of the chemical reactions occurring between TMA and PMMA (or the configurations of the TMA-PMMA complexes) was different from that observed in earlier cycles and what was observed in the low-temperature SIS.

Notably, the $A_{shifted\;C=O}$ values at the end of TMA purge steps in the 2nd and subsequent cycles performed at 130 °C were in the order of a noise level due to the fast dissociation of TMA-PMMA complexes and the fast out-diffusion of TMA molecules. Even when the concentration of the TMA-PMMA complex remaining at the end of the TMA purge step in the 2nd and subsequent cycles was negligible, the AlO_x_H_y_ formation occurred in the later cycles, as confirmed by the increase in the Al mole fraction of this sample (~42%) in comparison to that of the low-temperature 1-cycle sample (~22%) (Figure 5a). This implies that there is an additional mechanism, rather than the formation of reversible TMA-PMMA complexes, that is responsible for the AlO_x_H_y_ synthesis in the later cycles at 130 °C, although the exact mechanisms could not be identified by the in situ FTIR spectroscopy study at present. Therefore, we propose that an ALD-like reaction occurred in the vicinity of AlO_x_ clusters, which were formed in the 1st cycle performed at 80 °C, as depicted in Figure 6f. The presence of surface -OH groups, as shown in Figure 6b, and the dominance of the -OH component in the Al 2p HRXPS spectrum, as shown in Figure 5d, corroborates our hypothesis regarding the ALD-like reaction, which results in the retention of hydroxyls during ALD of aluminum oxides at a low temperature.

## 4. Conclusions

In this work, we investigated changes in the chemical binding (or bonding) states of sorbed TMA molecules that contribute the most to oxide formation as a function of growth conditions via correlation of in situ FTIR spectroscopy and other ex situ characterization techniques. The changes in the major chemical species are associated with the variations in the rate-limiting step: the association/dissociation of the TMA-PMMA complex vs. the in-diffusion/out-diffusion of TMA. For example, at an elevated temperature and a long TMA purge, the out-diffusion of TMA in PMMA is a rate-limiting step, and freely diffusing TMA molecules, not being bound to a PMMA matrix, are the major species contributing to the aluminum oxide/hydroxide formation in the PMMA matrix. Finally, we proposed a two-temperature scheme for the efficient and uniform formation of aluminum oxides/hydroxides along the direction of PMMA film thickness. This was achieved by the ALD-like growth of aluminum oxides/hydroxides in high-temperature cycles near the pre-existing nuclei formed during the preceding low-temperature cycle.

## Figures and Tables

**Figure 1 sensors-22-06132-f001:**
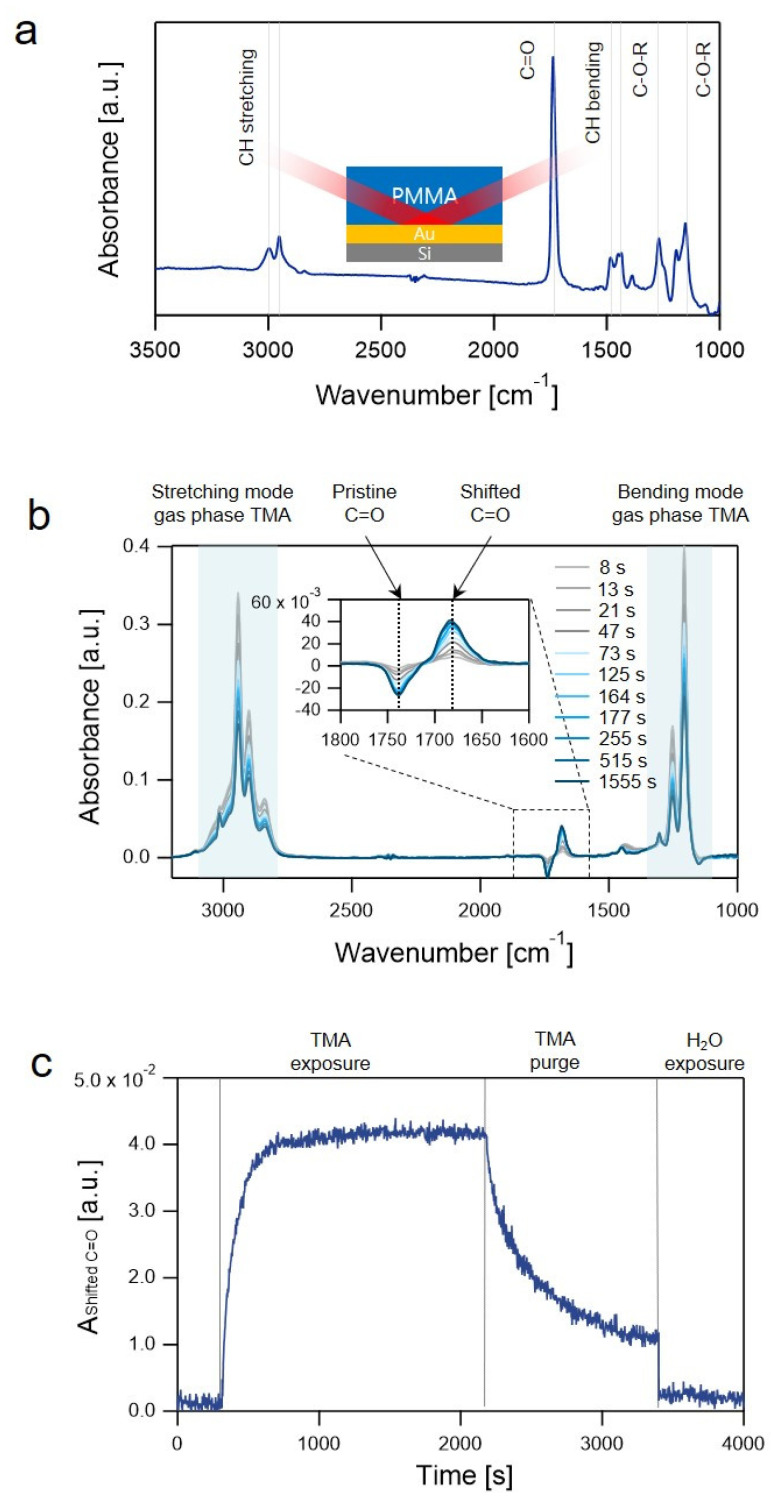
(**a**) Absorbance spectrum of the pristine PMMA/Au/Si sample prior to precursor exposure. The inset of (**a**) shows a schematic of the sample structure of the PMMA/Au/Si substrate; (**b**) absorbance spectra of PMMA/Au/Si collected at different times during the TMA exposure step at 80 °C; and (**c**) time-lapse absorbance spectrum of a shifted C=O band at 1670 cm^−1^ in (**b**) during the TMA dose-exposure-purge step and H_2_O dose-exposure step at 80 °C.

**Figure 2 sensors-22-06132-f002:**
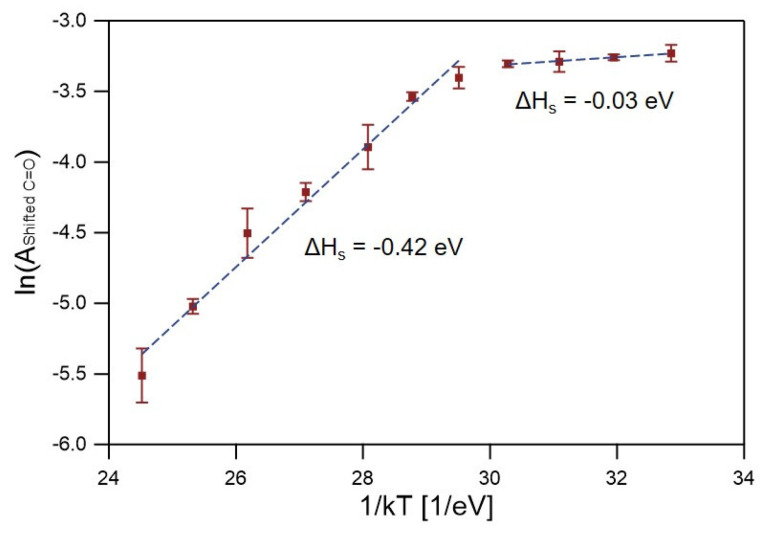
Saturated absorbance of the shifted C=O band as a function of temperature. Error bars represent standard deviations calculated using five different measurements (*n* = 5).

**Figure 3 sensors-22-06132-f003:**
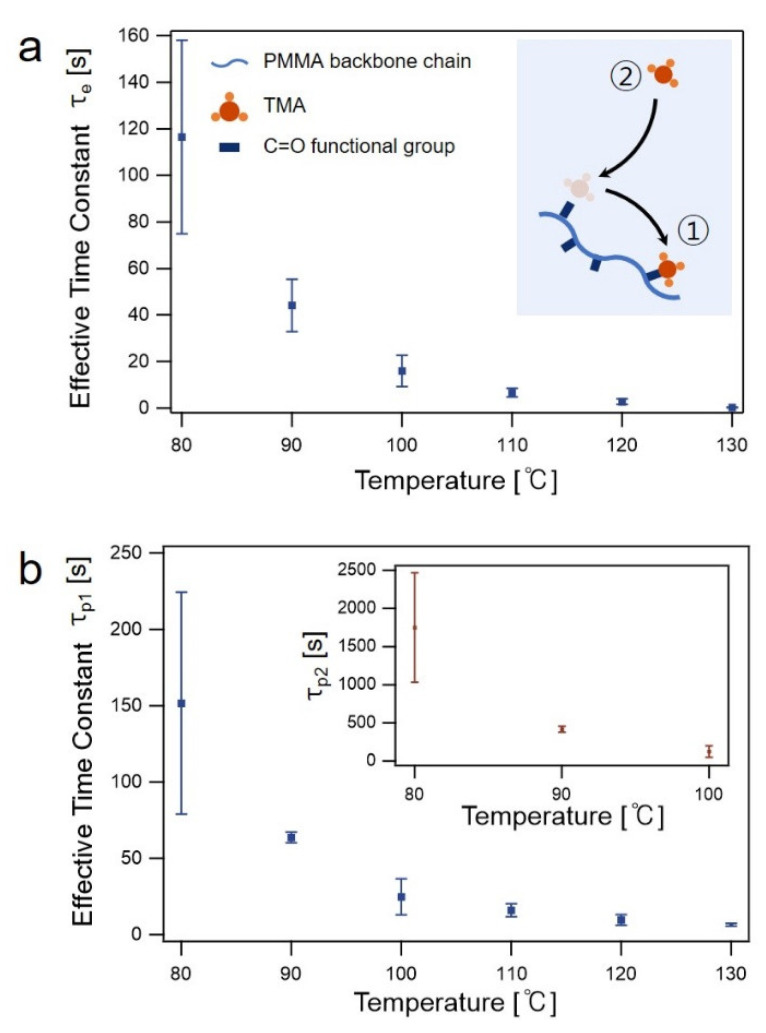
Effective time constants measured in (**a**) TMA exposure step ($\tau_{e}$); and (**b**) purge step ($\tau_{p1}$) as a function of temperature. Inset of (**b**) shows the effective time constant ($\tau_{p2}$) at a longer time scale. Error bars represent standard deviation values calculated using five different measurements.

**Figure 4 sensors-22-06132-f004:**
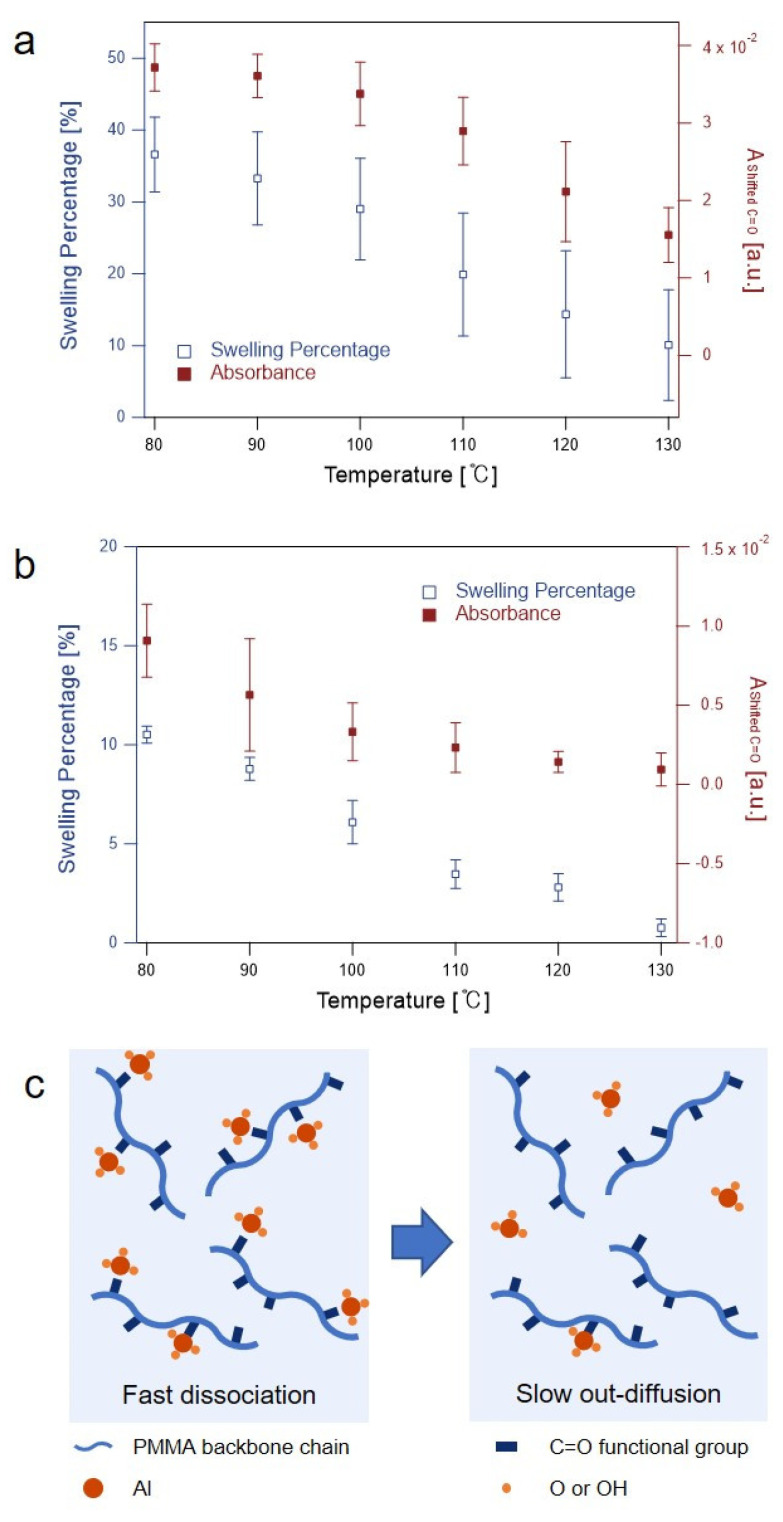
Comparisons of swelling percentage of the hybrid film after one SIS cycle and $A_{shifted\;C=O}$ measured at the end of each TMA exposure step followed by TMA purge step: (**a**) short purge time (5 s); and (**b**) long purge time (10 × *τ_p_*_1_). The error bars in (**a**,**b**) were obtained from five different measurements; and (**c**) schematic illustration of the fast dissociation of TMA-PMMA complexes and the out-diffusion of unbound TMA molecules at a high temperature for a long purge time.

**Figure 5 sensors-22-06132-f005:**
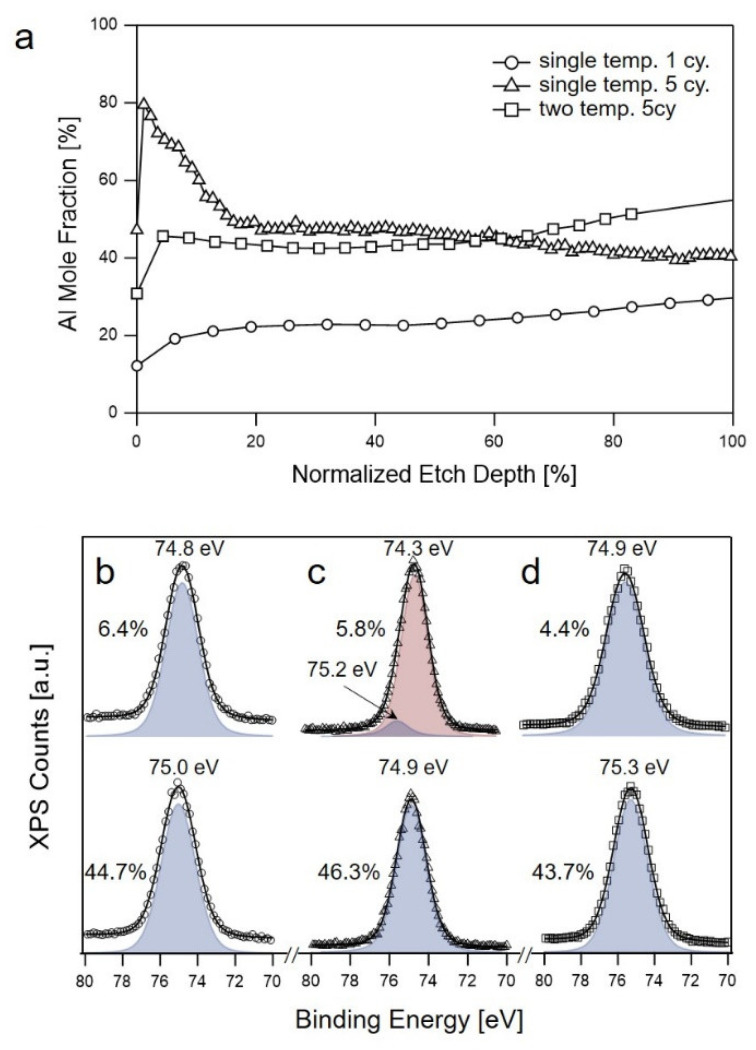
(**a**) XPS depth profiles of three samples prepared under different conditions: single-temperature SIS of 1 cycle at 80 °C; single-temperature SIS of 5 cycles at 80 °C; two-temperature SIS of 5 cycles-1st cycle at 80 °C and the last 4 cycles at 130 °C. The 100% etch time corresponds to the position of the interface between the PMMA film and the Si substrate. Al 2p HRXPS collected at the two etch times of the same three samples prepared under the following conditions: (**b**) single-temperature SIS of 1 cycle; (**c**) single-temperature SIS of 5 cycles at 80 °C; and (**d**) two-temperatures SIS of 5 cycles. The percentage values in (**b**–**d**) correspond to the percent etch time.

**Figure 6 sensors-22-06132-f006:**
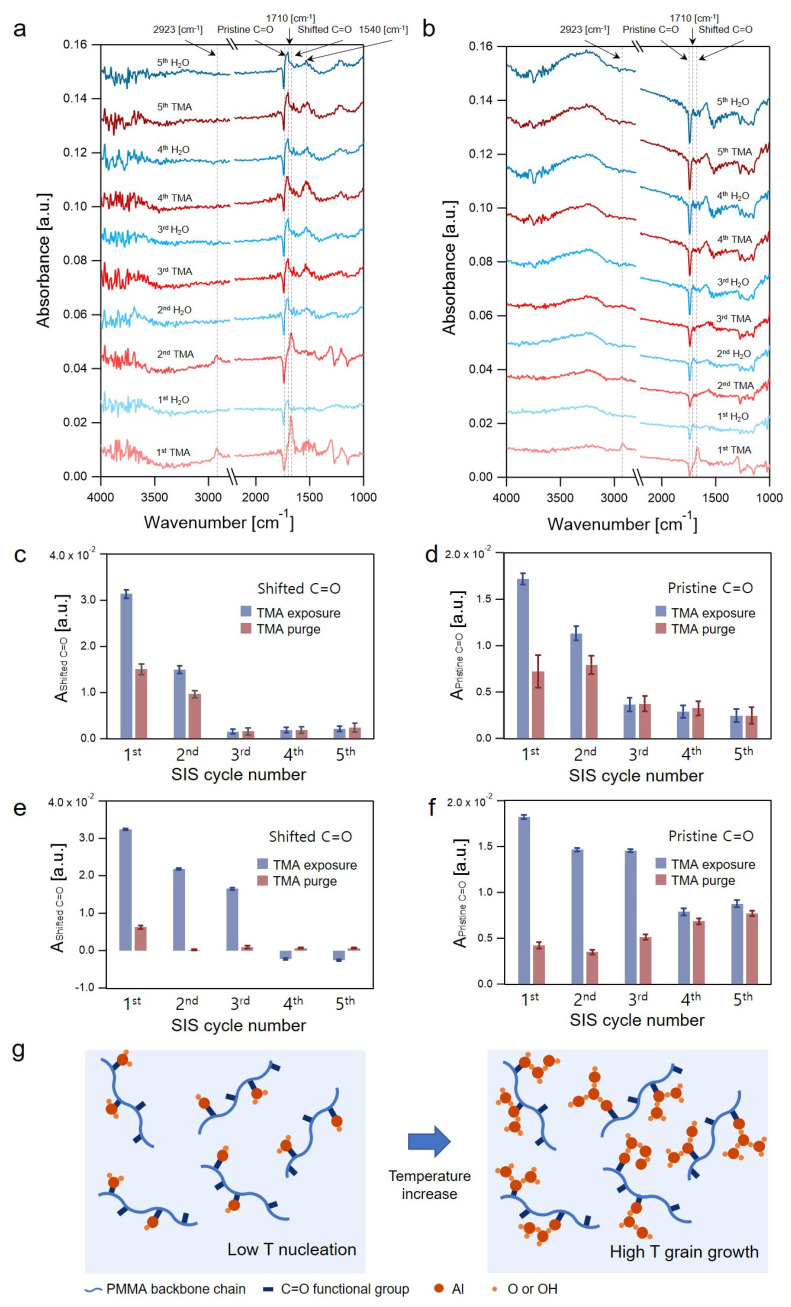
Absorbance spectra collected at the end of the TMA and H_2_O purge steps in: (**a**) single-temperature SIS of 5 cycles; and (**b**) two-temperature SIS of 5 cycles. Variations in the absorbance of (**c**) the shifted C=O band; and (**d**) pristine C=O band collected at the end of TMA exposure and TMA purge steps as a function of the SIS cycle number in the single-temperature SIS of 5 cycles. Variations in the absorbance of: (**e**) the shifted C=O band; and (**f**) pristine C=O band collected at the end of TMA exposure and TMA purge steps as a function of the SIS cycle number in the two-temperature SIS of 5 cycles; and (**g**) schematic illustration of the ALD-like reaction near AlO_x_H_y_ clusters in the two-temperature SIS experiment.

## Data Availability

Not applicable.

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
