# Peer review of "Understanding Physicochemical Mechanisms of Sequential Infiltration Synthesis toward Rational Process Design for Uniform Incorporation of Metal Oxides"

_sensors, 2022, doi:10.3390/s22166132_

Round 1

Reviewer 1 Report

This manuscript by Ham et al reports on the understanding of the mechanisms involved in the process of sequential infiltration synthesis in order to gain control over it. In particular, in situ FTIR was used together with several ex situ analysis techniques to study the growth of aluminium oxides in PMMA using TMA as the metal precursor and water as co-reactant.

The manuscript is well written, the work presented is a systematic one, gives a complete and up to date bibliography, and conclusions are well supported by the results. However, some aspects need to be improved and/or clarified. In particular:

1.       It is mentioned in the title that this study aims at the morphology tuning and also in the abstract it is stated that the objective is to obtain precise control over the morphology but I am not sure whether this is correct. In fact I would say there is no information about the morphology (form, shape, structure) but, in any case, about the uniformity of metal oxide incorporation

2.       Was the annealing step really needed in order to remove toluene?

3.       What is the roughness of the PMMA films?

4.       Figure 1a should be described in the experimental section, better than in the results section.

5.       The same for the explanation about the use of gold to enhance the reflectivity. It would be more appropriate to give this information in the experimental section.

6.       In figure 4, error bars are obtained from how many measurements?

Reviewer 2 Report

The description of experimental methods specially atomic layer deposition reactor with schematic figure, characterization methods, results, discussion should be more if this paper was submitted for a regular issue without page limit. Specially figures  need appropriate discussion and deteriorating the interesting results even though I recommend strongly the revision of manuscript. A notation list is necessary.

Round 2

Reviewer 1 Report

There are no additional comments after implementing/answering the changes/questions made in the previous revision

Reviewer 2 Report

Improved than previous version